

# Genome-wide identification, characterization and expression analysis of the DUF668 gene family in tomato

Hui Li[*], Tingrui Zou[*], Shuisen Chen and Ming Zhong

Key Laboratory of Agricultural Biotechnology of Liaoning Province, College of Biosciences and Biotechnology, Shenyang Agriculture University, Shenyang, China
[*] These authors contributed equally to this work.

## ABSTRACT

The domain of unknown function 668 (DUF668) is a gene family that may play a key role in plant growth and development as well as in responding to adversity coercion stresses. However, the DUF668 gene family has not yet been well identified and characterized in tomato. In this study, a total of nine putative *SlDUF668* genes were identified in tomato, distributed on six chromosomes. Phylogenetic analyses revealed that SlDUF668 proteins were classified into two major groups. Members within the same group largely displayed analogous gene structure and conserved motif compositions. Several *cis*-elements were exhibited in the upstream sequences of the *SlDUF668* genes, including elements implicated in plant growth and development processes, abiotic stress and hormone responses. Further, the study assessed the expression patterns of the SlDUF668 gene family in various tomato tissues, five plant hormones treatments, three abiotic stresses using qRT-PCR. The *SlDUF668* genes expressed ubiquitously in various tissues, and five genes (*SlDUF668-04*, *SlDUF668-06*, *SlDUF668-07*, *SlDUF668-08* and *SlDUF668-09*) showed tissue specificity. And *SlDUF668* genes responded to abiotic stresses such as salt, drought and cold to varying degrees. Overall, our study provided a base for the tomato DUF668 gene family and laid a foundation for further understanding the functional characteristics of *DUF668* genes in tomato plants.

## INTRODUCTION

The Pfam database classifies and names the domains of unknown function (DUF) using a combination of "DUF" and numbers, such as DUF1 and DUF2 (*Mistry et al., 2021*; *Bateman, Coggill & Finn, 2010*; *Schultz et al., 1998*). Chris Ponting first proposed the naming scheme of DUF in 1998 by adding DUF1 and DUF2 to the SMART database. Subsequently, DUF1 and DUF2 were renamed based on their featured peptides as the GGDEF (PF00990) domain and EAL (PF00990) domain, respectively (*Bateman, Coggill & Finn, 2010*). These domains have two distinct characteristics: a relatively conservative amino acid sequence and a protein domain with an unidentified function (*Bateman, Coggill & Finn, 2010*). The number of DUF superfamily members has rapidly increased in recent

Corresponding authors
Shuisen Chen,
shuisenchen@syau.edu.cn
Ming Zhong,
mingzhong@syau.edu.cn

years due to the sequencing of genomes from a large number of species. As of 2010, the entire family expanded to DUF2607 (DUF1-DUF2607). (*Bateman, Coggill & Finn, 2010*). Currently, the Pfam database (version 35.0) contains 19,632 families, with approximately 24% (4,795 out of 19,632) being composed of DUF families (*Mistry et al., 2021*).

Proteins containing the DUF domain are known to have a significant impact on the growth and development of plants. *Arabidopsis thaliana*, for instance, has several DUFs that regulate the biosynthesis of its plant cell wall, including DUF266, DUF231, DUF246, DUF1218, and DUF579 (*Parsons et al., 2012*; *Yuan et al., 2013*; *Oikawa et al., 2010*; *Mewalal et al., 2016*; *Urbanowicz et al., 2012*). Some DUFs, such as DUF640 and DUF827, control the development of chloroplasts and plant growth (*Zhao et al., 2004*; *Kodama, Suetsugu & Wada, 2011*). Additionally, DUFs like DUF828, DUF966, DUF668, DUF642, and DUF761 regulate the growth of roots (*Shen et al., 2019*; *Zhao et al., 2021*; *Gao et al., 2012*; *Salazar-Iribe et al., 2016*; *Zhang, Zhang & Huang, 2019*). The DUF640 family primarily contributes to the regulation of flower development (*Li et al., 2012*), while the DUF784 and DUF1216 families are involved in pollen development (*Huang et al., 2008*; *Jones-Rhoades, Borevitz & Preuss, 2007*).

In addition to controlling plant growth and development, DUFs have been discovered to play a role in plant stress responses. For instance, the *AtRDUF1* gene (*DUF1117*) positively regulates responses to salt stress in *Arabidopsis thaliana*, and suppressing the expression of both *AtRDUF1* and *AtRDUF2* decreases tolerance to drought stress mediated by ABA in *Arabidopsis* (*Kim, Ryu & Kim, 2012*). *ESK1* (AT3g55990), a member of the DUF231 gene family, acts as a negative regulator of cold acclimation (*Xin et al., 2007*). Overexpressing *GmCBSDUF3* (*DUF21*) enhances tolerance to drought and salt stress in *Arabidopsis* (*Hao et al., 2021*). The *AhDGR2* gene from *Amaranthus hypochondriacus* encodes a DUF642 protein that is involved in salt and ABA hypersensitivity in *Arabidopsis* (*Palmeros-Suárez et al., 2017*). The *OsDSR2* gene, which encodes a DUF966 protein, negatively regulates salt stress, simulated drought stress, and ABA signaling in rice (*Luo et al., 2013*). *OsSIDP361*, a *DUF1644* gene from rice, improves tolerance to drought and salinity (*Li et al., 2016*). Overexpressing the *OsDUF946.4* and *OsSIDP366* genes enhances tolerance to high salt and drought in rice (*Li et al., 2017*; *Guo et al., 2016*). *OsSGL* provides increased drought tolerance in transgenic rice and *Arabidopsis* (*Cui et al., 2016*). Other *DUF* genes that are related to abiotic stress in rice include *OsDSR2* (*DUF966*) (*Luo et al., 2013*) and *OsDUF810.7* (*Li et al., 2018*). Some *TaDUF966* genes are induced by salt stress in wheat (*Triticum aestivum* L.), and the role of the *TaDUF966-9B* gene in salt stress has been confirmed (*Zhou et al., 2020*). Overexpressing the salt-inducing gene *TaSRHP*, a *DUF581* gene from wheat, enhances resistance to salt and drought stress in *Arabidopsis thaliana* (*Hou et al., 2013*). Overexpressing *AmDUF1517*, a gene responsive to cold stress, significantly improves tolerance to various stresses in transgenic cotton (*Hao et al., 2018*). Some *GmDUF668* genes significantly respond to salt stress in soybean (*Zaynab et al., 2023*). Some ZoDUF668 genes were upregulated under cold stress in ginger (*Han et al., 2024*). Four genes (*IbDUF668-6*, *7*, *11* and *13*) of sweet potato were significantly upregulated under ABA, drought and NaCl stress (*Liu et al., 2023*). Other *DUF668* genes that are related to abiotic stress in cotton include *Gh_DUF668-05*, *Gh_DUF668-08*, *Gh_DUF668-11*, *Gh_DUF668-23*

and *Gh_DUF668-28* (*Zhao et al., 2021*). All *OsDUF668* genes respond to drought (*Zhong et al., 2019*).

The tomato is a widely cultivated vegetable crop that has significance in global agricultural production (*Gruber, 2017*). But, its growth and yield are greatly impacted by abiotic stressors (*Solankey et al., 2004*). The potential significance of *DUF668* family genes in plant stress resistance has been demonstrated, but research on this gene family has been limited to *Arabidopsis thaliana*, rice, cotton, and sweet potato. However, the function, classification, and evolution of this gene family in tomatoes have not been thoroughly investigated (*Zhong et al., 2019*; *Zhao et al., 2021*; *Liu et al., 2023*; *Zaynab et al., 2023*; *Han et al., 2024*). In this study, DUF668 gene family was systematically identified, and bioinformatic analysis was performed based on the whole tomato genome. We comprehensively analyzed physicochemical properties, chromosomal location, evolutionary relationships, gene structure, protein motifs, *cis*-acting elements of promoter and expression pattern of nine *DUF668* genes in tomato. The results of this study will provide a reference for further research on their possible functions in development and abiotic stress in the future.

## MATERIALS AND METHODS

### Plant materials and treatments

Tomato plants (*Solanum lycopersicum* Mill. cv. Ailsa Craig) were grown in a growth chamber in soil under a 25 °C/16 h in light condition, 20 °C/8 h in dark condition and 60% relative humidity. For drought and salt stress, 4-week-old plants grown in soil were watered with 20% PEG6000 mass/volume fraction and 200 mM NaCl solution and grown at normal room temperature, respectively. For low-temperature treatment, plants were placed in a light incubator at a temperature of 4 °C for 24 h. For phytohormone treatments, solution of 100 μM ABA, 50 μM MeJA and 50 μM SA solutions were sprayed onto tomato plants. Tomato leaves of different treatments were collected randomly after 0, 1.5, 3, 9, 12 and 24 h time courses. Roots, stems, young leaves, shoot apexes, young flower buds, anthesis flowers, green fruits and ripening fruits were collected for the analysis of *SlDUF668* expression levels. All samples were rapidly placed in liquid nitrogen and then stored in a refrigerator at −80 °C refrigerator. All samples were treated with three sets of replicates and four technical repetitions, and each replicate consisted of ten seedlings.

### Identification of *DUF668* genes in tomato genomes

In order to identify the tomato *DUF668* genes, the whole-genome was downloaded from the tomato Genome Database (https://solgenomics.net/) (*Fernandez-Pozo et al., 2015*). The hidden Markov model (HMM) profile of the DUF668 protein (PF05003) was obtained from the Pfam database, and the potential gene family members of *DUF668* was obtained *via* the HMMER search (https://www.ebi.ac.uk/Tools/hmmer/) (*Wheeler & Eddy, 2013*), with an *E*-value ($\leq 1e^{-5}$). After removing redundant and incomplete sequences, the conserved domain architectures of the acquired sequences were further confirmed by the Simple Modular Architecture Research Tool (SMART) database (http://smart.embl-heidelberg.de) (*Letunic, Khedkar & Bork, 2021*) and Pfam database (http://pfam.xfam.org/) (*Mistry et al.,*

*2021*). The number of amino acid residues, molecular weight (MW), and theoretical isoelectric point (pI) of each tomato SlDUF668 protein were predicted by ExPASy software (https://www.expasy.org/) (*Mariethoz et al., 2018*). The subcellular localizations of tomato SlDUF668s were predicted by the WoLF PSORT program (https://wolfpsort.hgc.jp/) (*Horton et al., 2007*).

## Phylogenetic and collinearity analysis of the SlDUF668 gene family

The full-length amino acid sequences of six DUF668 proteins of *Arabidopsis thaliana*, twelve DUF668 proteins of rice, thirty-two DUF668 proteins of cotton, fourteen DUF668 proteins of sweet potato and nine DUF668 proteins of tomato were aligned by ClustalW with default settings (*Larkin et al., 2007*). Subsequently, a phylogenetic tree of DUF668 was constructed with MEGA 11.0 using the neighbor-joining (NJ) methods, and the validated bootstrap value was set to 1,000 using the proximity method (*Kumar et al., 2008*). The results were edited and beautified with the online tool Evolview (https://evolgenius.info/) (*Subramanian et al., 2019*). BlastP was performed between the different species, and MCScanX was used to search all collinearity gene pairs and finally visualized through Circos (http://circos.ca/) (*Krzywinski et al., 2009*; *Lavigne et al., 2008*; *Wang et al., 2012*). Non-synonymous (Ka) and synonymous (Ks) substitutions rates of the duplicated gene pair were obtained from PGDD to evaluate the evolutionary selection (*Lee et al., 2013*).

## Chromosome localization, gene structure and protein motif analysis of *SlDUF668* genes

Chromosome position information of the SlDUF668 gene family members was obtained from the annotation file of the ITAG 4.0 tomato genome and chromosome localization was visualized using Mapchart software (*Voorrips, 2002*). The exon/intron arrangement of *SlDUF668* genes was analysed and visualized with TBtools (*Chen et al., 2020*). MEME software v5.0.5 (https://meme-suite.org/meme/tools/meme) was used to identify conserved motifs with the default parameters (*Bailey et al., 2009*). The conserved motifs were visualized using TBtools (*Chen et al., 2020*). All genes were renamed based on their positions across chromosomes.

## Promoter *cis*-acting element analysis

Sequences 2,000 bp upstream of start codon of the *SlDUF668* genes were downloaded from the tomato genomes and submitted to the PlantCARE database (http://bioinformatics.psb.ugent.be/webtools/plantcare/html/) for promoter *cis*-acting regulatory element screening. The possible cis-acting elements was visualized with TBtools after the statistical screening (*Lescot et al., 2002*).

## RNA extraction and reverse transcription-quantitative PCR (RT-qPCR) analysis

Total RNA was extracted using Plant RNA Extraction kit (Tiangen, Beijing, China) and the concentration of the isolated RNA was measured by a NanoDrop 1000 spectrophotometer (Thermo Fisher Scientific). The cDNA was synthesized by SMART kit (Takara), with 2 μg of RNA from each sample according to the manufacturer's protocol. The qRT-PCR assay

**Table 1  Characteristics of the putative *DUF668* gene family in tomato.**

| Gene name | Gene ID | Chr | Open reading frame/bp | Protein length/aa | Relative Molecular weight (r)/KDa | Theoretical isoelectric point (pI) | Subcellular localization |
|---|---|---|---|---|---|---|---|
| SlDUF668-01 | Solyc03g007090.1 | Chr03 | 1437 | 478 | 54.29 | 9.48 | Chloroplast |
| SlDUF668-02 | Solyc04g025210.4 | Chr04 | 1890 | 629 | 70.90 | 9.06 | Cytoplasm |
| SlDUF668-03 | Solyc04g081510.4 | Chr04 | 1803 | 600 | 67.13 | 9.43 | Cytoplasm |
| SlDUF668-04 | Solyc06g065460.2 | Chr06 | 1935 | 644 | 71.89 | 9.14 | Chloroplast |
| SlDUF668-05 | Solyc09g008930.4 | Chr09 | 1746 | 581 | 65.86 | 9.10 | Chloroplast |
| SlDUF668-06 | Solyc09g089640.2 | Chr09 | 1764 | 587 | 65.91 | 8.38 | Nucleus |
| SlDUF668-07 | Solyc10g084880.3 | Chr10 | 1407 | 468 | 53.20 | 8.99 | Nucleus |
| SlDUF668-08 | Solyc11g007660.1 | Chr11 | 1812 | 603 | 67.29 | 9.17 | Chloroplast |
| SlDUF668-09 | Solyc11g017000.2 | Chr11 | 1935 | 644 | 71.42 | 7.77 | Chloroplast |

was performed with a CFX96TM real-time fluorescent qPCR system (Bio-Rad, USA) using SYBR Green kit (Tiangen, Beijing, China) under the following conditions: 95 °C 2 min, followed by 35 cycles of 95 °C 15 s, 60 °C 30 s, and 72 °C 15 s. The *EF1a* gene (AY905538) was used as an internal reference (*Aoki et al., 2010*). Information about the primers for qRT-PCR were designed by Primer Premier 6.0 software and listed in Table S1. All the qRT-PCR analyses set three biological replicates and three technical repetitions for each treatment. The relative expression of the target gene was computed using the $2^{-\Delta\Delta Ct}$ method (*Livak & Schmittgen, 2001*) and visualized as heatmaps by TBtools. SPSS 20.0 was used to analyze the relative expressions and Origin 9.0 was used to complete the histogram of relative expression.

## RESULTS

### Identification and physicochemical properties of the DUF668 gene family members in tomato

A total of nine DUF668 protein sequences were characterized using the bioinformatics approach in tomato. We named the nine *DUF668* genes as *SlDUF668-01* to *SlDUF668-09* according to the order in which genes are distributed on chromosomes in tomato (Table 1). The open reading frame (ORF) length of *SlDUF668s* ranged from 1,407 to 1,935 bp, with the amino acid length of SlDUF668 proteins ranged from 581 to 644. The protein molecular weights ranged from 53.20 to 71.89 kDa, and the theoretical isoelectric point (pI) ranged from 7.77 to 9.48. The prediction of subcellular localization results showed that most of the genes were distributed in the chloroplast, and a few genes were distributed in the nucleus and cytoplasm.

Chromosomal location indicated that nine *SlDUF668* genes were distributed on six chromosomes (Chr03, Chr04, Chr06, Chr09, Chr10 and Chr11) of tomato (Fig. 1). Two putative *SlDUF668* genes were located on Chr04, Chr09 and Chr11, and the rest of the chromosomes contain only one *DUF668* gene. Although Chr09 has the longest length, it contains only one *SlDUF668* gene. These results suggest that there is no significant positive correlation between chromosome length and gene number.

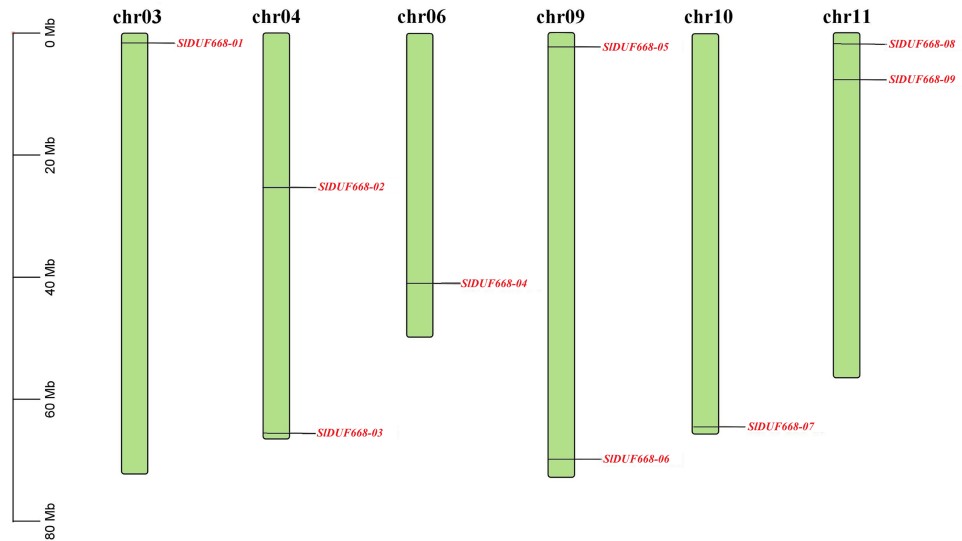

**Figure 1** **Chromosome distribution of the *Sl DUF668* genes on tomato genome.** The chromosome number is indicated at the top of each chromosome, and red letters represent *Sl DUF668* genes. The left scale indicates the size of each chromosome.

## Phylogenetic analysis, gene duplication and synteny analysis of DUF668 gene family in tomato

In order to extend our understanding of the relationship of evolution on SlDUF668, we constructed tree of the nine SlDUF668s, six *Arabidopsis thaliana* DUF668s, twelve rice DUF668s, thirty-two cotton DUF668s and fourteen sweet potato DUF668s by using the protein sequences (Table S2). The tree showed that these DUF668 proteins could be classified into two groups based on their groupings with *Arabidopsis thaliana* and rice DUF668 proteins (Fig. 2). Group 1 contained five tomato DUF668 proteins, three *Arabidopsis* DUF668 proteins, six rice DUF668 proteins, twenty-two cotton DUF668 proteins, and nine sweet potato DUF668 proteins. Group 2 contained four tomato DUF668 proteins, three *Arabidopsis* DUF668 proteins, six rice DUF668 proteins, ten cotton DUF668 proteins and five sweet potato DUF668 proteins.

To better study the evolution of the *SlDUF668* genes, we investigated the duplication events of *SlDUF668* family genes. The results showed that two segmental duplication events involving nine *SlDUF668* genes were identified, but no tandemly duplicated gene pairs were detected (Fig. 3, Table S3). In addition, the non-synonymous (Ka) and synonymous (Ks) substitution values and Ka/Ks ratios were conducted on these two identified duplicated *SlDUF668* gene pairs. The results showed that all two gene pairs have Ka/Ks values of less than 1 (Table S3), which suggests that the duplicated *SlDUF668* gene pairs underwent purifying selection in the course of evolution.

To further explore the evolutionary relationship of the SlDUF668 gene family, we analyzed the duplication events among five representative species, including four dicots (tomato, cotton, sweet potato and *Arabidopsis*) and one monocot (rice) (Fig. 4). We found that 18 *SlDUF668* genes were syntenic with the *DUF668* genes of *Arabidopsis thaliana*

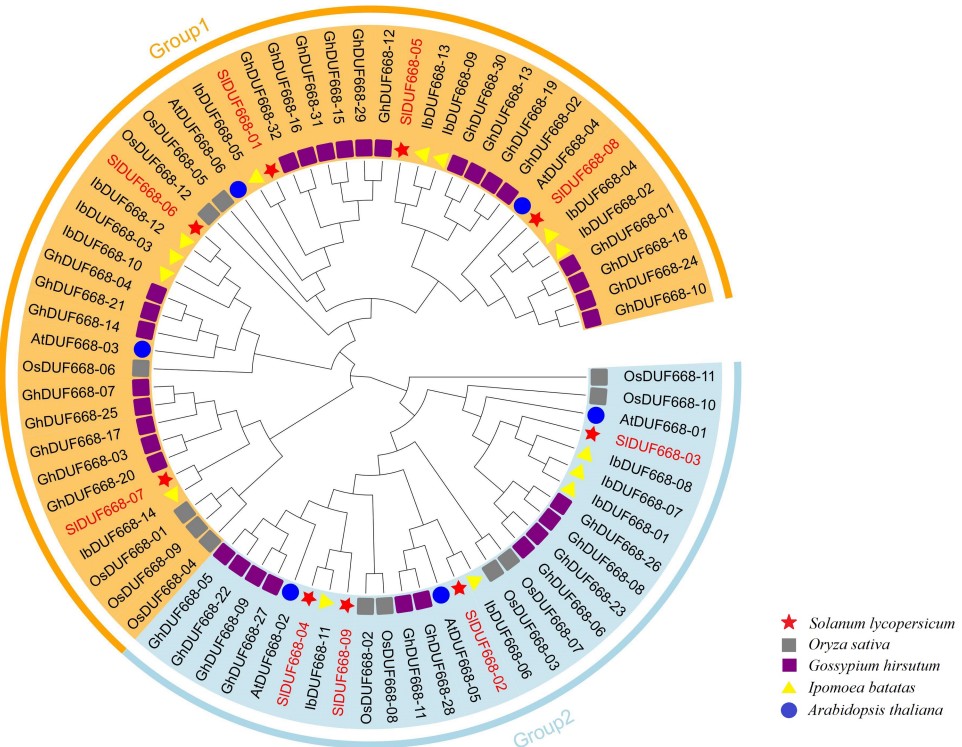

**Figure 2** Phylogenetic analysis of SlDUF668 in tomato, *Arabidopsis thaliana*, sweet potato, rice, cotton. All SlDUF668 members were classified into two groups, and different color blocks represent one group. The stars, circles, triangles, gray squareres and purple squares presented the DUF668 proteins from tomato, *Arabidopsis thaliana*, sweet potato, rice and cotton, respectively.

(three), sweet potato (two), cotton (12) and rice (one) (Table S4). Taken together, the syntenic gene pairs of *SlDUF668* were more presented in dicot than in monocot.

## Phylogenic tree, conserved motifs and gene structure of *SlDUF6688* gene

To explore further the phylogenetic relationship of SlDUF668 gene family, we constructed a neighbor-joining (NJ) phylogenetic tree based on the SlDUF668 protein sequences from tomato to fully analyze the conserved motif and gene structure (Fig. 5A). Ten putative conserved motifs were identified by MEME motif analysis, named motif 1–10 (Fig. 5B and Table S5). SlDUF668 protein in Group 2 had more motifs than Group 1. In addition, motif 1, 2, 3, 4, 5 and 7 were present in all the SlDUF668 proteins, suggesting that these motifs may be a conserved structure of the SlDUF668 family. Motif 8 and 10 were only found in Group 2, and motif 6 only existed in Group 1. In the investigation of the composition of the *SlDUF668* gene, we found that the Group 1 subgroups all have only one exon, while the Group 2 all contain 12 exons except *DUF668-02* (13 exons) (Fig. 5C). In conclusion, the motif constituent and exon-intron structures of the *DUF668* genes were very closely related with their phylogenetic relationships.
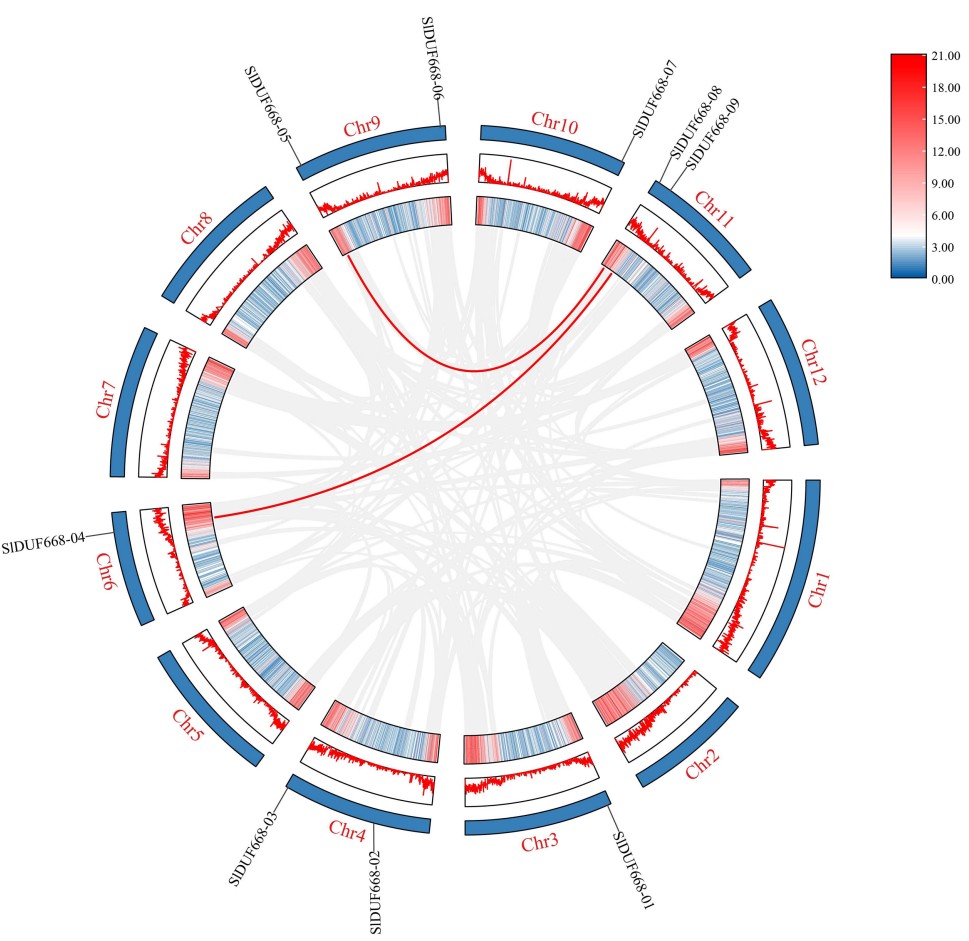

**Figure 3   Interchromosomal relationships of tomato *SlDUF668* genes.** The red line indicates the segmental duplication and the gray lines in the background represent the collinearity between the same genome.

## *Cis*-acting elements analysis of *SlDUF668* genes in tomato

There is a significant positive correlation between the response gene and *cis*-acting elements in gene promoter regions (*Walther, Brunnemann & Selbig, 2007*; *Abdullah et al., 2018*). Therefore, we analyzed the potential *cis*-acting elements of *SlDUF668* genes I promoter regions *via* the PlantCARE online website. The identified *cis*-acting elements could be mainly classified into three main categories: plant growth metabolism, plant hormones response and stress response (Fig. 6, Fig. S1). The predicted plant growth metabolism elements mainly included zein metabolism regulation (O2-site), meristem expression (CAT-box), endosperm-specific expression (AACA motif), root-specific expression (the motif I), circadian and palisade mesophyll cell differentiation (HD Zip1) and flavonoid biosynthesis (MBSI) elements. The predicted plant hormones response elements mainly included auxin (IAA), abscisic acid (ABA), gibberellin (GA), methyl jasmonate (MeJA) and salicylic acid (SA) elements. The predicted stress response elements mainly included WUN motif (wound-responsive), LTR (low-temperature response), MBS (drought-inducibility),

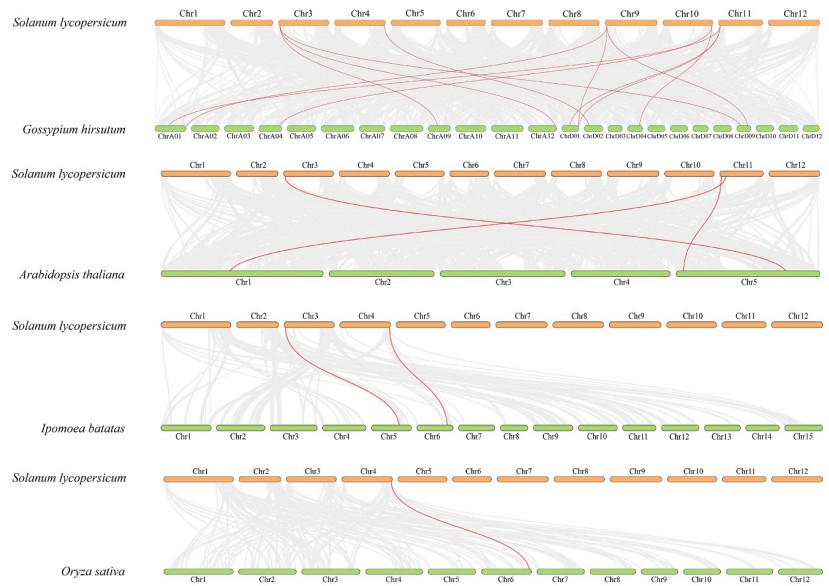

**Figure 4 Interspecific collinearity relationship between *SlDUF668* genes and DUF668s from *Arabidopsis thaliana*, sweet potato, rice, cotton.** The chromosomes of tomato are represented in orange, while those of *Arabidopsis thaliana*, sweet potato, rice and cotton are represented in green. The grey lines indicate the collinear blocks respective genomes, while the red lines represent the homologous gene pairs.

ARE (anaerobic induction), and TC-rich repeat elements (related to defense and stress response). The presence of these *cis*-acting elements in *SlDUF668* genes indicated their potential integral roles in different biological processes.

## Expression analysis of tomato *SlDUF668* genes in different tissues

To better understand the function of *SlDUF668* genes in tomato, we examined the expression patterns of the ten *SlDUF668* genes in different tissues using qRT-PCR, including roots, stems, young leaves, shoot apexes, young flower buds, anthesis flowers, green fruits and ripening fruits. The SlDUF668 gene family members were expressed in different tissues (Fig. 7), indicating they have diverse functions. *SlDUF668-04*, *SlDUF668-06*, *SlDUF668-07*, *SlDUF668-08* and *SlDUF668-09* were mainly expressed in anthesis flowers, *SlDUF668-06*, *SlDUF668-07*, *SlDUF668-08* and *SlDUF668-09* were mainly preferentially high-expressed in flower buds. *SlDUF668-08* and *SlDUF668-09* were high-expressed in shoot apexes, while *SlDUF668-04* and *SlDUF668-06* were highly expressed in ripening fruits, suggesting that *SlDUF668-04* and *SlDUF668-06* might play important roles in fruit ripening. *SlDUF668-08* were high-expressed in all tissues, indicating their diverse biological functions. In addition, all *SlDUF668* genes except *SlDUF668-07* were relatively low-expressed in roots.

## Expression profiles of *SlDUF668* genes under plant hormone

To explore the possible roles of *SlDUF668* genes in response to different hormonal treatments, we examined their expression levels under SA, MeJA, GA₃, IAA and ABA
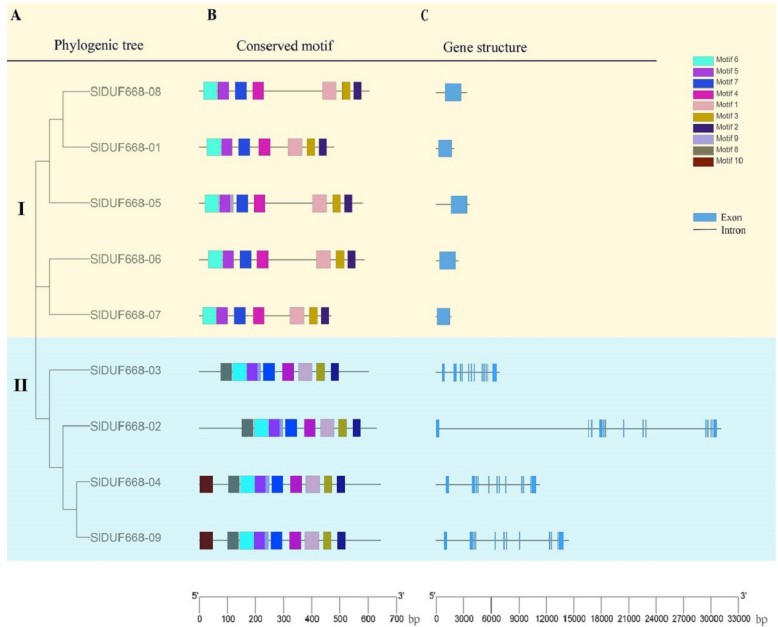

**Figure 5** **Phylogenetic tree, gene structures and protein motif analysis of DUF668 gene family in tomato.** (A) A phylogenetic tree of tomato DUF668 gene family. The blocks of different colors represented different groups. (B) Conserved domains of tomato DUF668 proteins. The colorful boxes represented different motifs. (C) Gene structures of tomato DUF668 genes. Green boxes represented exons and black lines represented introns. The clustering was performed according to the results of phylogenetic analysis.

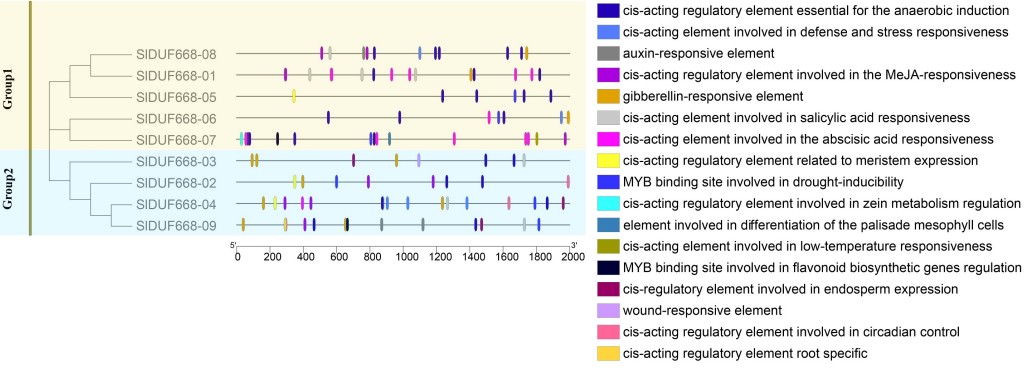

**Figure 6** **The *cis*-acting elements predication in the promoters of SlDUF668 gene family.** The different colors indicated the different *cis*-regulatory elements.

treatments by qRT-PCR (Fig. 8). During SA treatment, the expression levels of *SlDUF668-03*, *SlDUF668-04* and *SlDUF668-08* were up-regulated significantly at the 1.5 h and 3 h time point, respectively, and then showed a downward trend over time. After treatment with MeJA, the expression levels of *SlDUF668-04*, *SlDUF668-06* and *SlDUF668-08* was up-regulated considerably at 1.5 h and then showed a downward trend over time. Most of the *SlDUF668* genes were either only slightly induced or not affected by GA$_3$ treatment

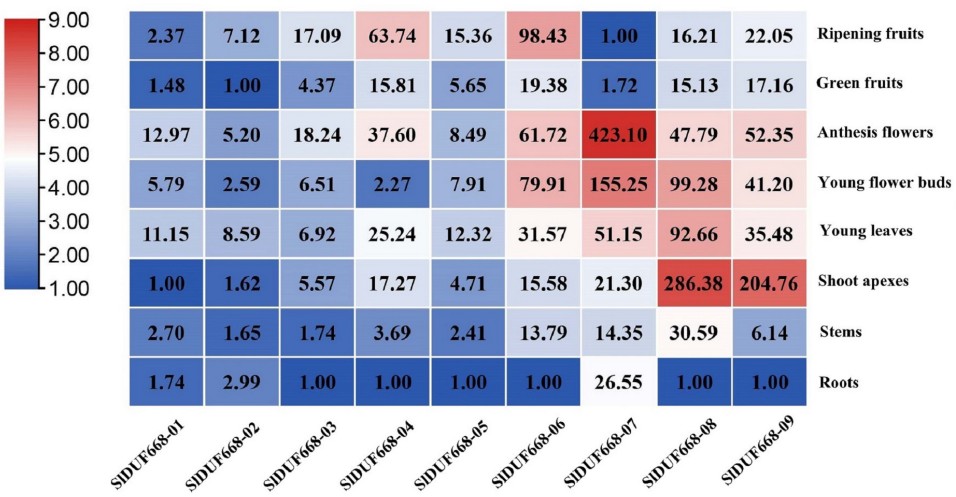

**Figure 7 Expression patterns of *SlDUF668* genes in eight tissues.** The genes were displayed at the bottom of each column and the tissues were labeled on the right. The heat map of the expression levels of *SlDUF668* genes in different tissues was calculated by the $2^{-\Delta\Delta CT}$ method and generated by TBtools. The color gradient (red/white/blue) indicates the gene expression level (high to low).

except *SlDUF668-01*, which was up-regulated at 1.5 h after GA₃ treatment compared to control. Expression of *SlDUF668-02*, *SlDUF668-03*, *SlDUF668-04*, *SlDUF668-06* and *SlDUF668-09* were elevated when applying with IAA. During ABA treatment, *SlDUF668-02* and *SlDUF668-03* were both obviously increased at 1.5 h compared with 0 h, then decreased at 24 h. In addition, the expression levels of *SlDUF668-06* and *SlDUF668-09* were significantly up-regulated at 3 h and 9 h, respectively, then decreased at 24 h. The expression levels of *SlDUF668-05* and *SlDUF668-07* were down-regulated from 1.5 to 24 h after treatment in comparison with 0 h. *SlDUF668-01* showed increasing expression only at 24 h after treatment compared to control.

## Expression of *SlDUF668* genes under abiotic stresses

To investigate the potential responsiveness of *SlDUF668* genes to abiotic stresses, the gene expression of *SlDUF668* genes under different abiotic stresses was analyzed using qRT-PCR (Fig. 9). After 24 h of cold stress treatment, the expression of three genes (*SlDUF668-01*, *-03* and *-05*,) significantly changed, suggesting that the expression of these three genes may be crucial in response to cold stress. The expression of four genes (*SlDUF668-01*, *-02*, *-06*, *-07* and *-09*) was induced and reached a maximum at 24 h, meanwhile *SlDUF668-07* showed strong up-regulation of over five times, indicating that these four genes might play a role in drought resistance in tomato. Under salt stress treatment, the expression of four genes (*SlDUF668-01*, *-02*, *-03* and *-05*) significantly changed, suggesting that these three may play an important part in salt. Additionally, some genes showed a trend of gradual increase after stress under environmental stresses, such as *SlDUF668-04*, *-06*, *-07*, *-08* and *-09* for NaCl, *SlDUF668-02*, *-04*, *-06*, *-08* and *-09* for cold and *SlDUF668-04* and *-05* for PEG stress. In summary, these results indicated that the *SlDUF668* family members may act as key roles in plant reactions to diverse abiotic challenges in tomato.

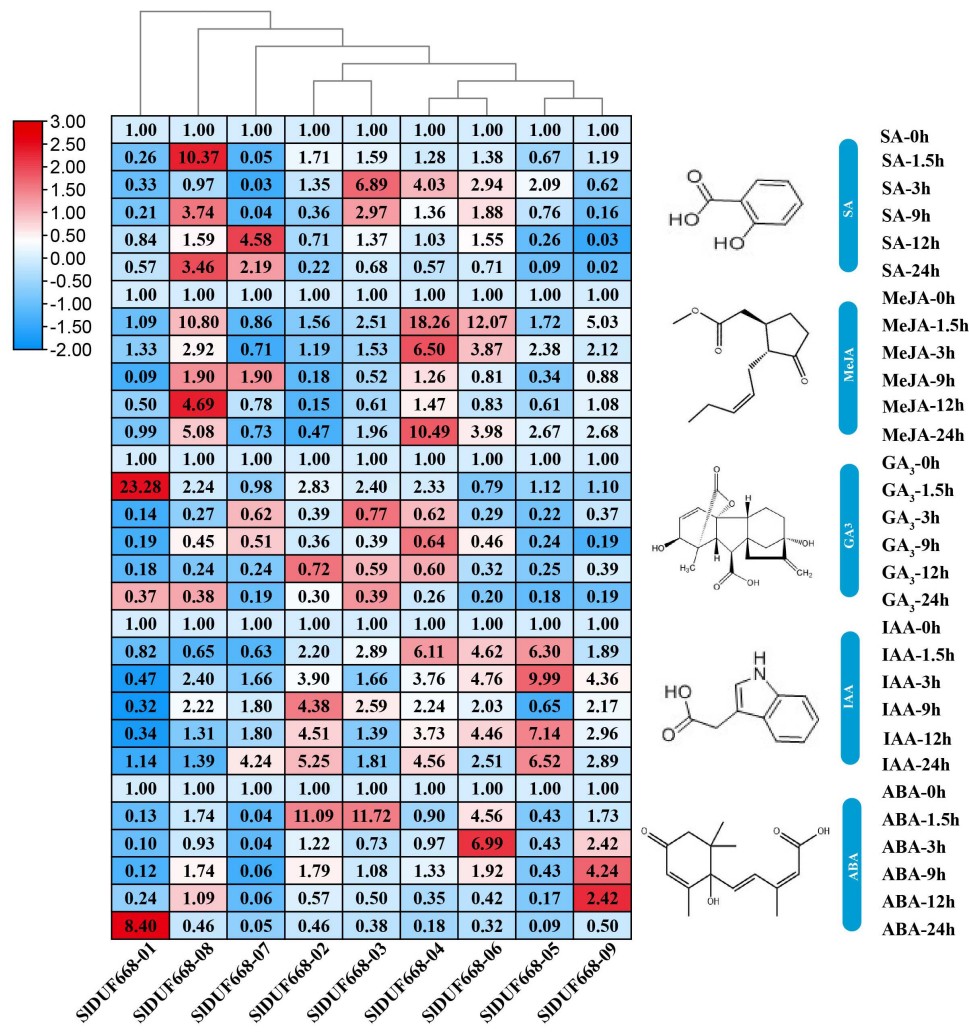

**Figure 8** **Expression profiles of *Sl DUF668s* under SA, MeJA, GA, IAA and ABA treatments.** SA, salicylic acid; MeJA, methyl jasmonate; GA, gibberellic acid; IAA, indole-3-acetic acid and ABA, abscisic acid. The relative expressions of SlDUF668 genes were detected by qRT-PCR after treated at five-time points (1.5, 3, 9, 12 and 24 h) were normalized to 0 h treatment. The fold changes values were calculated by the $2^{-\Delta\Delta CT}$ method and log2 and visualized by heat map. Red and blue colors indicate up-regulation and down-regulated expression levels to the control respectively. Value for each time point represents the mean of three biological replicates.

## DISCUSSION

The DUF668 gene family is significant in plant growth, development, and stress responses (*Zhong et al., 2019*; *Zhao et al., 2021*; *Liu et al., 2023*; *Han et al., 2024*; *Zaynab et al., 2023*). Several plant genomes, including *Arabidopsis thaliana*, rice, cotton, and sweet potato, have undergone genome-wide identification of the DUF668 gene family. The release of the high-quality tomato genome SL4.0 in 2019 (*Hosmani et al., 2019*) presented an opportunity to study the entire genome of the DUF668 gene family in tomatoes, which has remained unexplored to date. In this study, we discovered nine *SlDUF668* genes in

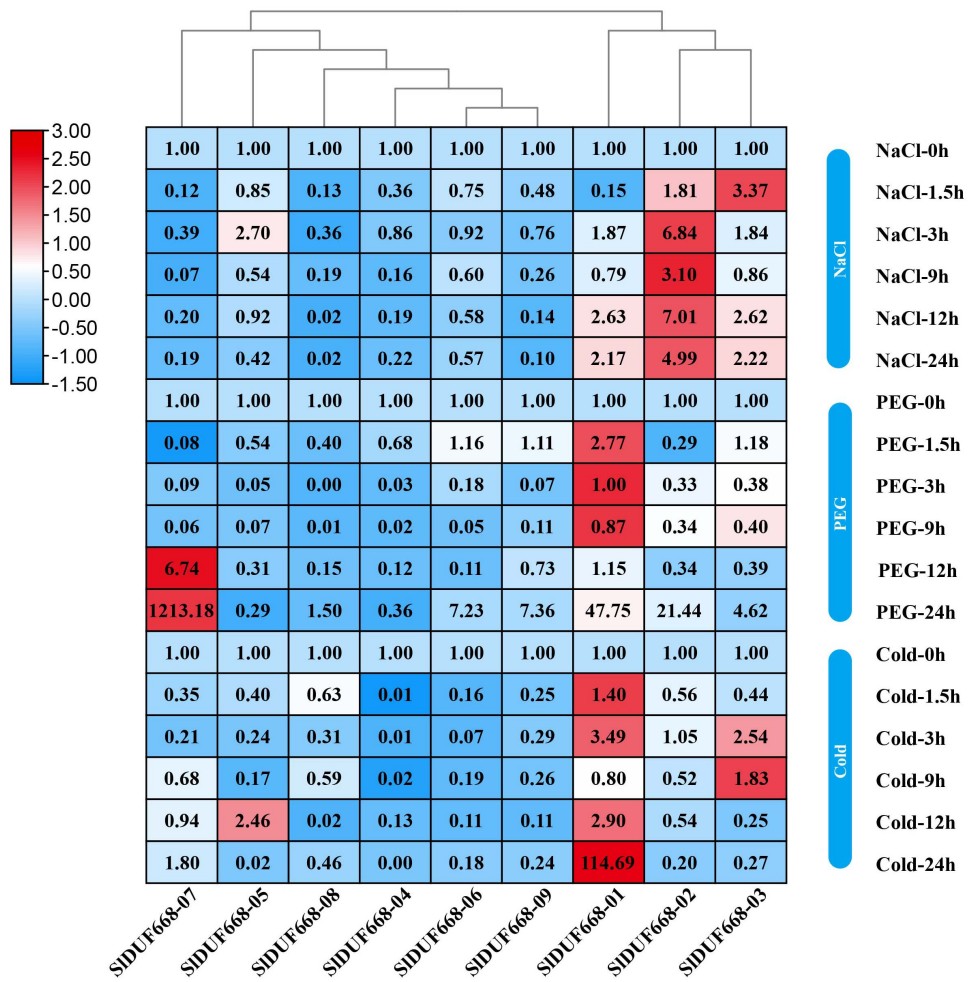

**Figure 9  Expression changes of *SlDUF668s* under abiotic stress conditions.** The relative expressions of *SlDUF668* genes were detected by qRT-PCR after treated at five-time points (1.5, 3, 9, 12 and 24 h) were normalized to 0 h treatment. The fold changes values were calculated by the $2^{-\Delta\Delta CT}$ method and log2 and visualized by heat map. Red and blue colors indicate up-regulation and down-regulated expression levels to the control respectively. Value for each time point represents the mean of three biological replicates.

the tomato genome, distributed across six chromosomes (Table 1, Fig. 1). Interestingly, the number of SlDUF668 gene family members in tomatoes exceeded that of *Arabidopsis* but fell short of the numbers observed in rice, cotton, and sweet potato. These findings indicated that the number of *DUF668* genes is not directly related to the genome size of a species. Furthermore, the subcellular localization prediction indicated that a majority of the *SlDUF668* genes were situated in the chloroplast (Table 1), and similar results were observed in the investigations of *SlDUF668* genes in rice (*Zhong et al., 2019*), implying that these genes may play a role in regulating the structure of the chloroplast. Analysis of the phylogenetic relationship demonstrated that the SlDUF668 proteins could be categorized into two groups (Fig. 2), and no significant tandem duplication phenomenon was observed (Figs. 3 and 4), suggesting that tandem duplications occurblue less frequently during the

expansion and evolution of the DUF668 gene family. The *SlDUF668* genes in the same group exhibited notable similarities in terms of their gene structure and motif composition (Fig. 5), indicating that the groupings of *SlDUF668* genes were relatively reliable. Despite the smaller number of genes in Group 2, the *DUF668* gene in this group was longer than that in Group 1 and contained more motifs and exon structures, suggesting that SlDUF668 in Group 2 may possess more intricate functions. These findings were highly congruent with the *SlDUF668 genes* discovered in other species (*Zhong et al., 2019*; *Zhao et al., 2021*; *Liu et al., 2023*; *Han et al., 2024*; *Zaynab et al., 2023*).

There is significant evidence indicating that differences in gene activities are often linked to variations in promoter regions (*An, 1986*; *Lyu et al., 2015*). *Cis*-elements play a crucial role in plant regulatory networks, contributing to a deeper understanding of transcriptional regulation and uncovering the functions of associated genes (*Hernandez-Garcia & Finer, 2014*). Predictive analysis of the *SlDUF668* family gene promoter revealed the existence of multiple elements involved in plant growth metabolism, plant hormones response and stress response (Fig. 6). Among them, plant hormones response and stress response elements were most widely presented, such as ABREs, MBS, LTR, WUN-motif, TGACG, CGTCA, GARE motif, etc. The MBS element that participate in the plant response to drought stress (*Zhang et al., 2012*), the TGACG motif and CGTCA motif in the MeJA hormone response, the GARE motif in the gibberellin response and ABREs mainly in the ABA hormone response. These hormonal response processes can be indirectly participate in plant responses to abiotic stresses (*Rouster et al., 1997*; *Imtiaz et al., 2015*; *Yoshida et al., 2010*; *Huang et al., 2017*; *Qi et al., 2018*). These findings suggest that the *SlDUF668* genes can enhance the tomato's responses to abiotic stress, especially drought, salt and cold stress. This is consistent with the results of the expression analysis of *SlDUF668* genes under drought, salt and cold stress in this study.

Gene expression is associated with gene function (*Zhu et al., 2021*). The expression levels of the *SlDUF668* genes were observed in different tissues. The findings showed that the *SlDUF668* genes were ubiquitously expressed in different tissues (Fig. 6), indicating that they may play necessary functions in different tissues. *SlDUF668-04*, *-06*, *-07*, *-08* and *-09* exhibited highly expressed in some specific tissues (Fig. 7), suggesting that the expression of these *SlDUF668* genes in tomato is tissue specific. Abiotic stresses such as low temperature, salinity and drought play an important role in plant growth and yield. Many studies have also shown that hormones are required for plants to responses to abiotic and biotic stresses. Therefore, a comprehensive analysis of *SlDUF668* genes under abiotic stresses and hormone treatments was performed in tomato. The results showed that some *SlDUF668* genes were significantly up-regulated in SA, MeJA, IAA and ABA treatments (Fig. 8), implying that these genes may play a key role in the hormone signaling pathway. Additionally, the *SlDUF668* genes were shown to have different expression levels under salt, drought and cold stresses, and the expression patterns of some genes distributed in the same group had similar expression trends (Fig. 9). For instance, *SlDUF668-01*, *-03* and *-05* positively responded to cold, indicating that they may enhance tomato resistance to cold. Similarly, *SlDUF668-01*, *-02*, *-06*, *-07* and *-09* were found to significantly up-regulated in response to drought stress, suggesting that these genes might may enhance tomato

resistance to drought. In addition, *SlDUF668-01*, *-02*, *-03* and *-05* positively responded to NaCl, suggesting that these genes might involve in the regulation of salt-induced stress. Overall, these comprehensive results served as a starting point for further research on their roles in tomato growth and development as well as environmental stress responses.

## CONCLUSIONS

In the current study, a total of nine *SlDUF668* genes were identified genome-wide in the tomato whole genome. Gene family analyses were conducted to investigate their physicochemical properties, chromosomal locations, phylogenetic relationships, gene architectures, conserved motifs analysis, cis-acting elements and expression patterns. According to phylogenetic and collinearity analyses, the nine *SlDUF668* genes were clustered into two groups. They were unevenly distributed on six chromosomes. Plant growth metabolism elements, plant hormones response elements and stress response elements were identified in the *SlDUF668* promoter. Expression analysis showed that *SlDUF668* genes had specific expression in different tissues and were widely involved in tolerance of abiotic stress in tomato. Taken together, we provide a valuable references to further understand the function of this gene family in tomato.

### Funding

This work was supported by the National Natural Science Foundation of China (Grant No. 31070448) and the Liaoning Province '2021 Special Project of Central Government Guiding local scientific technology development' (2021JH6/10500164). The funders had no role in study design, data collection and analysis, decision to publish, or preparation of the manuscript.

### Grant Disclosures

The following grant information was disclosed by the authors:
The National Natural Science Foundation of China: 31070448.
the Liaoning Province '2021 Special Project of Central Government Guiding local scientific technology development': 2021JH6/10500164.

### Competing Interests

The authors declare there are no competing interests.

### Author Contributions

- Hui Li conceived and designed the experiments, analyzed the data, prepared figures and/or tables, authored or reviewed drafts of the article, and approved the final draft.
- Tingrui Zou performed the experiments, analyzed the data, prepared figures and/or tables, authored or reviewed drafts of the article, and approved the final draft.
- Shuisen Chen performed the experiments, analyzed the data, prepared figures and/or tables, and approved the final draft.
- Ming Zhong conceived and designed the experiments, authored or reviewed drafts of the article, and approved the final draft.

## Data Availability

The raw measurements are available in the Supplementary Files.

## Supplemental Information

Supplemental information for this article can be found online at http://dx.doi.org/10.7717/peerj.17537#supplemental-information.

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
