# Peer review of "Genome-wide identification, characterization and expression analysis of the DUF668 gene family in tomato"

_PeerJ, doi:10.7717/peerj.17537_

## Round 0.1 · original submission · Minor Revisions

The authors need to include all suggestions given by reviewers.

**Language Note:** PeerJ staff have identified that the English language needs to be improved. When you prepare your next revision, please either (i) have a colleague who is proficient in English and familiar with the subject matter review your manuscript, or (ii) contact a professional editing service to review your manuscript. PeerJ can provide language editing services - you can contact us at [email protected] for pricing (be sure to provide your manuscript number and title). – PeerJ Staff

·

Basic reporting

Line 14: Change “identification” to “identified”.
Line 97: deleted the repeated word “and”.

Line 105: change the tense of the sentence (“were” instead of “should”).
Line 216: Add a space between “group” and “1”.
Line 259: the time point “3 h” is repeated.
Line 297: Instead of stating that the results indicate a direct relation between the number of genes and the size of the species, it would be more accurate to use the word "suggest" instead of "indicate" to convey the evidence found with an implication of the results.

For sentences in lines 332 and 333, please include the proper citations.

Line 338: please correct the sentence (“can were”).
Line 343: please correct the sentence (“might they may”).
Line 345: please correct the sentence (“might involved”).
Line 350: word misspelled (“physiochemical”).
Line 358: Please replace "for" with "to".
Figure 2: Replace “represents” with “represent”.
Figure 3: Eliminate the repeated word “the”.
Figure 5: The word “group” should be plural.

Experimental design

Line 103: Was there a scientific basis for choosing those specific time points to collect tomato leaves, or was it random?

Validity of the findings

In my humble opinion, this manuscript serves as a solid groundwork for future research, and the extensive range of the experiments opens up possibilities for more specific and focused experimentation.

Additional comments

Figure 7: Could the authors please clarify the meaning of the number "52%" that is enclosed in the green circle?

Reviewer 2 ·

Basic reporting

The paper is well-written overall, with only a few, minor spelling and/or grammatical errors (as noted/highlighted in the PDF). I have also requested some references to be added for the section where DUF668 genes are discussed in other crops (lines 85-87 in Page 7 of the manuscript PDF). I have also asked for a clarification on what the author wants to convey in lines 46-47 (page 6) of the manuscript.

Experimental design

No comment

Validity of the findings

I would have liked to see the functional similarities of the 2 distinct phylogenetic groups across the different plants explored further, especially since this group of genes has been sufficiently characterized in those plants.

Annotated reviews are not available for download in order to protect the identity of reviewers who chose to remain anonymous.

Reviewer 3 ·

Basic reporting

The manuscript entitled “Genome-wide identification, characterization and expression analysis of the DUF668 gene family in tomato” by Hui Li et al. focuses on the DUF668 gene family in tomato and its potential role in plant growth, development and stress responses. The study identified 9 putative SlDUF668 genes in tomato, distributed on six chromosomes, and classified them into two major groups based on phylogenetic analyses. The study also explored the gene structure, conserved motifs, and cis-elements associated with the SlDUF668 genes. Expression patterns of the SlDUF668 gene family in various tomato tissues, under different plant hormone treatments and abiotic stresses, were assessed using qRT-PCR. The findings revealed tissue-specific expression pattern and responses to stress conditions among the SlDUF668 genes, providing a foundation for understanding the functional characteristics of DUF668 genes in tomato plants.
The manuscript is written in a clear and logical way, however, there are a few points that need clarification or revision.

Experimental design

NA

Validity of the findings

1. Figure1 figure label, please check the alignment of chromosome number.
2. Figure2 legend, please rewrite for an improved grammar and clarity. “and different color blocks represents one group. The stars, circles, triangles, stars, gray squareres and purple squares presented the DUF668 proteins from tomato, Arabidopsis thaliana, sweet potato, respectively.”
3. Figure 3, “gray lines..represents” should be “gray lines..represent”. It’s better to provide scale details for the legend.
4. Figure 4, “lines represents” should be “lines represent”. Please provide a new figure with improved resolution.
5. Figure 5, “different group” should be “different groups”. The author has mentioned (A), (B) and (C) in the legend while none was labeled in the figure, it’s better to correct for improved clarity.
6. Figure 7, what’s the 52% on the right bottom of the figure? Please also provide the details for normalization of the expression level.
7. Figure 8, please rewrite the caption of figure 8 for an improved grammar and clarity.

Additional comments

1. “SlDUF688 genes” and “SlDUF688s genes” were both used through the manuscript, it’s better to be consistent.
2. Line 106: suggest “ -80 °C refrigerator” to be freezer.
3. Line 136: “was” should be “were”.
4. Line 163: “RT-qPCR” should be “qRT-PCR”.
5. Line 279, -07 should be SlDUF688-07.

Reviewer 4 ·

Basic reporting

In this study, author used bioinformatics method to identify the 9 SlDUF668 genes
in tomato. In addition, author found that SlDUF668s gene not only participate in growth
and development, as well as can play a crutial role in abiotic response of stress. The
methods and results are acceptable. There are some essential problems should be
addressed by authors, which are listed below

Experimental design

fine

Validity of the findings

fine

Additional comments

In this study, author used bioinformatics method to identify the 9 SlDUF668 genes
in tomato. In addition, author found that SlDUF668s gene not only participate in growth
and development, as well as can play a crutial role in abiotic response of stress. The
methods and results are acceptable. There are some essential problems should be
addressed by authors, which are listed below.
1. The abstract section needs to be further condensed and concise, and some very simple
conclusions do not need to be included in the abstract.
2. The manuscript has the problem of multiple positive italics, such as gene names, and
cis need italics, gene family names do not need italics.
3. Please note that 1-9 requires English words, not Arabic numerals. The article contains
several related errors. For instance, Line 15, 126, 127, 171, 172, 179, 189, 193-196,
198, 201, 276, 277, 280, 281, 282 and or so.
4. In the introduction, it is recommended to add how the DUF668 gene responds to
biotic and abiotic stress in rice, sweet tomato, cotton, wheat, soybean, and ginger.
5. L161-162, it should be stated why the EF1a gene was selected for an internal
reference. In addition, normality test and variance analysis were performed on the data
of RT-qPCR.
6. In the discussion, it is advisable to streamline the discussion regarding the results
obtained from bioinformatics analysis, emphasize on the mechanism of how SlDUF668
gene family responds to abiotic stress.
7. L315-322, the discussion of cis-element is not sufficiently exemplified and there are
too many conclusive descriptions. It is suggested to elucidate the mechanism of ciselement regulation of SlDUF668 in response to abiotic stress.
8. L333-334, the discussion of gene expression lacks previous studies. Please read the
literature of TaDUF668.
9. Some Figures and Tables should be modified.
In Figure 1, the gene names in the figure need to be italicized.
In Figure 2, add a legend corresponding to the species to the diagram
In Figure 5, horizontal axis missing units "bp" and "aa", suggest supplementing Intron
labels.In Figure 7, 8, and 9, please place the formula (log2(TPM+1)) to the left of the color
scale. What does the icon (52%) in the bottom right corner of Figure 7 mean?

Reviewer 5 ·

Basic reporting

Li and Zou et al. conducted a systematic identification of DUF68 family genes in the tomato genome and described the functional characteristics of the DUF668 gene family, encompassing physicochemical properties, chromosomal location, evolutionary relationships, gene structure, cis-acting regulatory elements, and expression patterns. This study provided a comprehensive overview of the roles played by the DUF668 gene family in tomato.

Experimental design

In the phylogenetic analysis of the SlDUF gene family, it is essential to select the longest isoform of the protein for this analysis. However, the authors did not provide clarification on this matter. It is necessary to verify whether the longest isoform was indeed chosen. Please revise and provide relevant descriptions in the Methods section.

Please add a description of the clustering of columns in Figures 8 and 9, and provide the details of data visualization.

Validity of the findings

In the Methods section, it is suggested to perform a coexpression analysis for the SIDUF668 genes examined in Figures 7, 8, and 9. The resulting cluster tree will be incorporated into the figures to facilitate understanding of the genes with similar expression patterns.

Additional comments

In the Discussion section, although the sequence of cis-elements remains conserved across different species, significant functional variations may arise depending on the activation of the transcription-factor binding motif. A paragraph will be suggested to discuss the correlation between the transcription-factor occupancy of these cis-elements and the transcriptional activation of target genes across distinct species, highlighting the roles of gene regulation in the evolutionary divergence.

---

## Round 0.2 · accepted · Accept

The revised manuscript is good to accept after English editing.

Reviewer 2 ·

Basic reporting

The authors have addressed all the clarifications I had in my original review. A few suggested changes in language:

Line 46-47: please rephrase the sentence about DUF2607, not sure what it means here
Line 95: change “gene family were” to “gene family was”
Line 139: Change “The BlastP” to “BlastP”
Line 140: Change “the MCScanX” to “MCScanX”
Line 147: change to “chromosome localization was visualized”
Line 149: change “The MEME software” to “MEME software”
Line 197: change to “tree of the nine SlDUF668s”

Experimental design

The experimental design is sound, and appropriate protocols were used whenever necessary.

Validity of the findings

The findings are novel, valid and relevant to the field. The conclusions made were appropriate.

Additional comments

No comments.

Reviewer 3 ·

Basic reporting

Based on revision of manuscript, authors successfully incorporated the point to point wise through the manuscript as suggested comments so that I would like to strongly recommend that acceptance of manuscript in PeerJ for publication.

Experimental design

NA

Validity of the findings

NA

Additional comments

NA

Reviewer 5 ·

Basic reporting

NA

Experimental design

NA

Validity of the findings

NA

Additional comments

The authors addressed all questions thoroughly. This manuscript has undergone significant improvement. Therefore, I recommend accepting it.